# The Coordinated Interplay Between MMP13 and Pro-Migratory MMPs in Collective Cell Migration of Zebrafish Keratocytes

**DOI:** 10.3390/ijms262211192

**Published:** 2025-11-19

**Authors:** Chandana K. Uppalapati, Marquise A. Jeffery, Agnes S. Pascual, Elizabeth E. Hull, Kathryn J. Leyva

**Affiliations:** 1Department of Microbiology & Immunology, College of Graduate Studies, Midwestern University, 19555 N. 59th Avenue, Glendale, AZ 85308, USA; cuppal@midwestern.edu; 2Biomedical Sciences Program, College of Graduate Studies, Midwestern University, 19555 N. 59th Avenue, Glendale, AZ 85308, USAapascu@midwestern.edu (A.S.P.); ehullx@midwestern.edu (E.E.H.)

**Keywords:** wound healing, collective cell migration, matrix metalloproteinases, MMP13, MMP14, MMP inhibitors, zebrafish keratocytes

## Abstract

Collective cell migration (CCM) is a coordinated process involving cell–cell and cell–environment interactions occurring in many physiological systems, including development, wound healing, and metastasis. Using zebrafish keratocytes as a wound healing model provides a unique system to investigate the interplay of matrix metalloproteinases (MMPs) in CCM. MMPs play an important role in CCM as they generate bioactive molecules that regulate proliferation, differentiation, angiogenesis, apoptosis, and cell migration. Secreted as pro-enzymes, MMPs must be activated, frequently by another MMP. As a group, MMPs have been reported to have a pro-migratory role during CCM, yet our data reveal that one MMP, MMP13, is not pro-migratory. Treatment of keratocytes with recombinant MMP13 resulted in a dose-dependent decrease in migration, reduced MMP13 activity, and increased *MMP9* mRNA expression. Treatment with an MMP13-specific inhibitor resulted in a dose-dependent increase in migration with no change in the rate of cellular proliferation, an increase in total MMP activity, and increased *MMP2* mRNA expression. Similarly, inhibition of MMP14 also resulted in a significant, dose-dependent decrease in migration. However, MMP14 inhibition resulted in both an increase in *MMP2* mRNA expression and a decrease in *MMP9* mRNA expression. The increase in MMP2 and/or MMP9 activity was observed on gel zymography for both treatments. Our data support the hypothesis that MMP13 is anti-migratory while MMP2, MMP9 and MMP14 have a pro-migratory effect on zebrafish keratocytes. Taken together, our results outline a novel inhibitory role for MMP regulation of CCM that has implications for many other processes in multicellular organisms.

## 1. Introduction

Collective cell migration (CCM) is defined as the movement of a group of cells in a distinct unit, referred to as a cell sheet [1,2]. Several investigators have defined the nature of collectively migrating cells, with cells at the front of the migrating edge being termed “leader cells” [1,2,3]. This definition of leader cells, in conjunction with Raftopoulo and Hall’s further characterization of follower cells and coordinated movements [3], allows for a supportive understanding of the complex mechanisms involved in collective cell migration that are occurring during a wounding event [1,2,4,5].

The migration of cells as a collective unit is most certainly regulated by several different signaling pathways; earlier studies elucidated the importance of stretch-activated calcium channels regulating calcium-dependent cortical contractions with microtubule rearrangements in the leading edge while maintaining cell–cell junctions during migration [3,6,7]. As expected, many signaling molecules have been identified that influence collective migration of cells, including growth factors, cytokines, chemokines, external stressors (e.g., wounding), calcium, and matrix metalloproteinases (MMPs) in several physiological systems [4,5,8,9,10,11,12,13,14,15,16,17,18,19]. MMPs are zinc-dependent endopeptidases that degrade extracellular matrix (ECM) proteins, facilitating removal of damaged collagen and promoting migration in several physiological systems [10,19,20,21,22,23,24,25,26,27,28,29,30,31,32]. The specific processes occurring during CCM is multifaceted and includes buildup and breakdown of cell–cell junctions, morphological and cytoskeletal changes in leader vs. follower cells, calcium–dependent contractions driving forward propulsion at the leading edge with concomitant retraction of cells at the trailing edge, and MMP degradation of extracellular matrix proteins. These processes must be tightly regulated in a coordinated fashion for the cell sheet to collectively migrate forward as a unit [3,6,7].

In addition to ECM degradation, MMPs are known to cleave various substrates, generating additional, tissue-specific signaling molecules to regulate CCM and other physiological processes [19,27,30,31,33,34,35,36,37,38,39,40]. The roles of MMPs in research have primarily centered around their involvement in pathological conditions, such as wound healing in diabetic complications and metastatic cancer growth via ECM invasion and degradation. Recently, their roles in other models as they relate to collective cell migration have been studied—these select MMPs include MMP2, MMP9, MMP13 and MMP14 [16,31,32,41,42,43]. These studies, and others, have shown that CCM in the context of wound healing is an essential process in order to close off a wound from the external environment and require MMPs.

Zebrafish have been used as an animal model to study several human physiological and pathologic processes, such as embryonic development, epithelial-to-mesenchymal transition, cancer metastasis, and wound healing [12,14,19,41,42,43,44,45,46]. Zebrafish keratocytes provide a unique opportunity to study the reepithelialization phase of wound healing, as the lack of bleeding removes any interference from clotting factors, allowing a more controlled setting to examine the effects of MMPs, or other factors, on migration.

In our zebrafish wound healing model, MMP2 and MMP9 have been previously shown to increase the migration of keratocytes; subsequently we showed that MMP14 activity, which activates MMP2 and MMP9, is localized and highly active at the leading edge of zebrafish cell sheets [5,41]. Our earlier work showed that when MMP2, MMP9, and MMP14 are inhibited, cell migration is markedly reduced, supporting a pro-migratory role for MMP2, MMP9 and MMP14 and also supported by Seomun et al. [47]. However, inhibition of MMP13 resulted in a dose-dependent increase in cell migration, providing initial evidence that MMP13 may be acting as an anti-migratory MMP [16,41]. These initial findings appear to contradict what has been previously reported in the literature on MMP13 [25,43,48,49,50,51], warranting further study into the role of MMP13 in the context of zebrafish keratocyte CCM. This present study focuses on the role of MMP13 in influencing proliferation, gene expression, and ultimately, cellular migration that could all be playing a part in the dynamics occurring during a wound healing event. The goal of this research was to explore the coordinated interplay between MMP13 and three known pro-migratory MMPs (MMP2, MMP9, and MMP14) using a zebrafish explant culture system as a unique model for wound healing in multicellular organisms.

## 2. Results

### 2.1. Localization of MMP Activity in the Cell Sheet

We observed that, in untreated keratocytes, cells along the leading edge of the cell sheet (leader cells) had significantly higher MMP activity, as measured by an increase in fluorescence intensity, compared to cells within the cell sheet (follower cells) (*p* < 0.001; Figure 1). MMP activity was significantly reduced when the cells were treated with either broad-spectrum MMP inhibitor: NNGH (*p* < 0.0001) or Z-PLG-NHOH (*p* < 0.0001), as expected (Figure 1).

### 2.2. Effect of MMP13 and MMP14 on Collective Cell Migration

Our lab has previously published that treatment of keratocytes with MMP inhibitors (MMP2, MMP9, and broad-spectrum [total] MMP inhibitors) resulted in a dose-dependent decrease in cell sheet area, while treatment with the MMP13-specific inhibitor WAY170523 yielded a dose-dependent increase in cell sheet migration [5]. We set out to test the effect of recombinant MMP13 or inhibiting additional MMPs on cell sheet migration.

As predicted, addition of recombinant MMP13 resulted in a dose-dependent decrease in cell sheet migration for the doses tested (Figure 2A). Treatment of cells with the MMP14-specific inhibitor NSC405020 also resulted in a dose-dependent decrease in cell sheet migration (Figure 2C), with the highest dose tested yielding no measurable cell sheets. To determine if there was a synergistic relationship, we treated the cells with an MMP9/13 specific inhibitor (MMP9/13 I; Santa Cruz Biotechnology, Dallas, TX, USA). We observed essentially no change in cell sheet area, except for a slight increase at the highest dose tested (Figure 2B), which indicates that these two MMPs are likely serving opposing roles on CCM.

### 2.3. Effect of MMP13 and MMP14 on Cell Proliferation

To determine if increases in cell sheet area were due to migration, and not proliferation, we performed an EdU assay to measure changes in proliferation following treatment. Treatment with either recombinant MMP13 or the MMP13-specific inhibitor WAY170523 resulted in no change in proliferation after 24 h (Figure 3; Appendix A). Initial treatments using the MMP14-specific inhibitor NSC405020 resulted in very little sheet growth when treated with 100 µM of the MMP14 inhibitor, and essentially no sheets were measurable at the highest dose tested (500 µM). However, the lowest dose tested (10 µM) visually appeared similar to untreated keratocytes, so we hypothesize that MMP14 likely does not substantially affect cellular proliferation, if at all, to explain increases in cell sheet area.

### 2.4. FRET Assay of MMP Activity in Culture Supernatants

We treated each culture supernatant with APMA before assessing either MMP13 activity or total MMP activity, as this allowed assessment of the total amount of either MMP13 present or collective MMPs present, respectively, after 24 h of treatment. However, we did perform an initial experiment assessing MMP activity in both APMA-treated and APMA-untreated supernatants; results yielded a similar pattern of MMP activity over time (Appendix A), although APMA-treated supernatants did have ~15% higher activity than untreated cells, as expected.

Treatment of cells with either an MMP13-specific inhibitor, an MMP14-specific inhibitor, or a broad-spectrum (total) MMP inhibitor significantly reduced the MMP13 activity compared to untreated (control) cells (*p* < 0.001, *p* < 0.001, *p* < 0.01, respectively (Figure 4). Interestingly, treatment of cells with 65 nM MMP13 also significantly reduced MMP13 activity in culture supernatants after 24 h (*p* < 0.05). We also assessed the total MMP activity in the same culture supernatants to determine if there would be an overall increase or decrease in total MMP activity with respect to treatment. As predicted, we observed a significant decrease in total MMP activity between untreated cells and cells treated with a broad-spectrum (total) MMP inhibitor (*p* < 0.05; Figure 4). Interestingly, we observed a significant increase in total MMP activity between untreated cells and cells treated with the MMP13-specific inhibitor WAY170523 (*p* < 0.01).

### 2.5. Effect of Recombinant MMP13 and MMP Inhibitors on MMP mRNA Expression

The effect of recombinant MMP13 or one of three MMP inhibitors on mRNA expression of four MMP genes (*MMP2*, *MMP9*, *MMP13a* and *MMP14a*) was examined after 24 h of treatment (Figure 5). Cells treated with 65 nM of recombinant MMP13 showed a significant decrease in *MMP2* mRNA expression (*p* < 0.0001) but a significant increase in *MMP9* mRNA expression (*p* < 0.0001). There was no effect on mRNA expression of either *MMP13a* or *MMP14a* following treatment with recombinant MMP13. Interestingly, treatment with an MMP13-specific inhibitor attenuates this effect, bringing *MMP9* mRNA expression back to a similar level as untreated (control) cells (*p* > 0.05) (Figure 5).

Compared to untreated cells, inhibition of MMP14 results in a significant increase in *MMP2* mRNA expression (*p* < 0.0001) while inhibition of MMP13 results in a significant decrease in *MMP2* mRNA expression (*p* < 0.0001). Remarkably, we see a similar decrease in *MMP2* mRNA expression as when cells were treated with recombinant MMP13. Additionally, inhibition of MMP14 also resulted in a significant decrease in *MMP9* mRNA expression as well (*p* < 0.01) (Figure 5). Treatment of cells with a broad-spectrum (total) inhibitor did not affect mRNA expression of *MMP2*, *MMP9* or *MMP14a* but did significantly increase *MMP13a* mRNA expression compared to untreated cells (*p* < 0.0001; Figure 5).

### 2.6. MMP Gelatinase Activity

We assessed the gelatinase activity of MMPs using gelatin zymography by quantifying the clear bands in the stained gel, which represents degradation of gelatin. The bands, when aligned with the corresponding molecular weight marker, represent the presence of pro and active forms of MMP2 and MMP9 (Figure 6A,B). The zymogram results show that inhibition of MMP13 resulted in a significant increase in activity of MMP2 (*p* < 0.01; Figure 6C) and MMP9 (*p* < 0.05; Figure 6D), and the addition of 130 nM recombinant MMP13 resulted in a significant increase in only MMP9 activity (*p* < 0.01; Figure 6D) when compared to the untreated control. These results are consistent with our mRNA expression data (Figure 5). However, treatment with an MMP14-specific inhibitor caused a significant decrease in MMP9 activity (*p* < 0.05; Figure 6D) when compared to control, also consistent with *MMP9* mRNA expression data (Figure 5). In contrast to the zymography data, we observed a significant decrease in *MMP2* mRNA expression and a significant (*p* < 0.001) increase in *MMP9* mRNA expression after treatment with an MMP13-specific inhibitor (Figure 5).

## 3. Discussion

Our zebrafish wound healing model provides a valid and rigorous system to evaluate the role of MMPs on collective cell migration (CCM). McDonald et al. showed that 17.5% of the genome was differentially expressed between cells after 1 vs. 7 days in explant culture with many of these genes identified as playing a role in wound healing, inflammation, and motility [41]. Among the expressed biomarkers, McDonald et al. found a high expression of matrix metalloproteinases present during early stages of EMT in zebrafish keratocytes [5,41]. Since it has been established that zebrafish keratocytes undergo EMT, the zebrafish wound healing model system affords us an opportunity in which to examine the influence of MMPs on collective cell migration [2,5,13,41,52]. As our explant culture system involves removing and immediately plating zebrafish scales that have attached keratocytes, we can evaluate the migration of cells at time zero of the wounding event, without any influence from the natural extracellular matrix (ECM), allowing us to study the function of MMPs outside of the context of their role in ECM degradation. While it has been established that MMPs do, in fact, play a significant role in migration in other systems, little work has been done on MMPs in the context of CCM using an ex vivo wound healing model system. This work focused on elucidating the interplay of MMP13 with other MMPs to promote CCM of zebrafish keratocytes.

Most of the recent literature on MMPs and their role in migration has shown that, collectively, MMPs are considered to be pro-migratory. In our lab, we have reported that MMP2, MMP9, and MMP14 all exert a pro-migratory effect on collectively migrating cells, and that migrating cells with prominent lamellipodia express a higher level of MMP14 on the cell surface than cells within the migrating sheet [5,16,41]. We also reported that MMP13 negatively affected cell migration, as opposed to the pro-migratory role of MMP2, MMP9, and MMP14; MMP13 inhibition resulted in a significant, dose-dependent increase in migration [5]. These previous results laid the groundwork for further investigating the role of MMP13 being anti-migratory, through the coordinated interplay with pro-migratory MMPs or other pro-migratory signaling pathways.

We initially wanted to establish whether migrating keratocytes produce active MMPs and if MMP production was localized within the cell sheet. Using a DQ gel assay, we were able to demonstrate that active MMPs are not only present and active within the explant cultures, but that they are localized to the leading edge of the cell sheet (Figure 1). We compared untreated 24 h sheets with two broad-spectrum MMP inhibitors to assess degree and localization of MMP activity. We found that compared to control, the broad-spectrum inhibitor NNGH was not as effective as Z-PLG-NHOH at reducing MMP activity via relative fluorescence, yet both were statistically significant from control (Figure 1). Our data illustrate that MMP activity is localized and more pronounced in the leading edge of the cell sheet.

Since MMP13 inhibition resulted in a significant, dose-dependent increase in cell sheet migration, we wanted to determine if treating cells with active, recombinant MMP13 would lead to a corresponding decrease in migration. Consistent with other studies [5,48,49,50,51], we observed a significant, dose-dependent decrease in migration when cells were treated with active, recombinant MMP13 (Figure 2A), supporting our previous work. We conclude that MMP13 decreases CCM (Figure 2A) while inhibiting MMP13 results in an increase in CCM [5]. We chose to investigate the effects of MMP13 further by inhibiting MMP14 (also known as MT-MMP), known to be one of the primary MMPs activating MMP13. MMP14 is expressed on the surface of zebrafish keratocytes and is known to cleave and activate a variety of MMPs, including MMP13. As we hypothesized, inhibition of MMP14 using the MMP14-specific inhibitor NSC405020 resulted in a significant, dose-dependent decrease in migration (Figure 2C). As MMP14 does activate other MMPs, we cannot separate whether the decrease in migration is due to the diminished activity of other, pro-migratory MMPs, or if the reduced MMP14 activity is directly producing an anti-migratory effect. It is known that cells from any tissue involved in CCM normally upregulate MMP14 at the leading edge, and thus, increase activation of downstream secreted MMPs, such as MMP2, MMP13 and MMP1 [16,26,35,41,53]. Our data support the hypothesis that MMPs do play a direct role in influencing CCM.

Treatment of the cells with MMPs or MMP inhibitors that cause an increase in overall size of the cell sheet after 24 h of growth can be explained by: 1. either an increase in migration of existing cells, 2. by proliferation of existing cells that contributes to the size of the cell sheet, or 3. by a combination of these two. We wanted to determine if MMP13 has any effect of cellular proliferation, as it has been reported to do in other experimental systems [43,54]. In our hands, either inhibiting MMP13 or adding recombinant MMP13 did not significantly affect cellular proliferation more so than in untreated cells (Figure 3). Our initial data, obtained using one dose (10 μM) of an MMP14 inhibitor, also did not show any increase in cellular proliferation; higher doses tested had a substantial inhibitory effect on migration (Figure 2C) that prevented accurate assessment of cellular proliferation at these doses. However, given that cell sheet area was small or not measurable at the higher doses, any effect on cellular proliferation is likely minimal after 24 h, and certainly not contributing to any increases in cell sheet areas. We conclude that the increase in cell sheet area observed with MMP13 inhibition is not a result of stimulating cellular proliferation; MMP13 directly (or indirectly) affects cell migration independent of proliferation.

In order to determine what effect MMPs or MMP inhibitors have on the overall MMP activity level within the explant culture system, we investigated both total MMP activity as well as MMP13-specific activity within culture supernatants. Previous studies have concluded that MMPs are upregulated during the first three days of collective cell migration [5,16,41,55], but no studies have examined whether MMP13 specifically is accounting for, or influencing the degree of, overall MMP activity. Our FRET data showed that when cells were treated with an MMP13-specific inhibitor, there was a statistically significant increase in total MMP activity compared to both untreated cells and cells treated with a broad-spectrum (total) MMP inhibitor. These data support our hypothesis that MMP13 is likely functioning to inhibit or dampen CCM (Figure 2A), possibly by inhibiting production or activity of pro-migratory MMPs, as we observed when both MMP9 and MMP13 were inhibited (Figure 2B). When MMP13 activity alone was assessed, all treatments resulted in a reduction in MMP13 activity, including addition of recombinant MMP13. Surprisingly, treating cells with 65 nM of recombinant MMP13 for 24 h resulted in a decrease in MMP13 activity after 24 h (Figure 4), which was unexpected but does support the decreased migration observed when cells were treated with recombinant MMP13 (Figure 2A). There potentially is a more complex interplay of these MMPs that is occurring, or that the high dose (130 nM; 10-fold higher than the IC_50_ value) of exogenous MMP13 added inhibited the cells from producing endogenous MMP13. Additional studies are needed to distinguish between these, or other, hypotheses and warrants further investigation. Further, when MMP14 is inhibited, we also see a decrease in MMP13 activity (Figure 4), supporting previous data that MMP14 is likely responsible for cleaving and activating MMP13 in zebrafish keratocytes [5,16].

We next analyzed the effect of treating keratocytes with either recombinant MMP13 or one of the three MMP inhibitors on MMP gene expression. Treatment of the cells with an MMP14-specific inhibitor resulted in a significant increase in *MMP2* mRNA expression and a corresponding decrease in *MMP9* mRNA expression (Figure 5). Curiously, we observed a slight (~1.4-fold) increase in *MMP14a* mRNA expression as well. Although not statistically significant, it suggests that *MMP14a* gene activation may be occurring as the MMP14 activity declines, whether through changes in overall MMP receptor signaling or directly through the inhibition of the MMP14 receptor itself. Even more interesting is that we observed a significant decrease in *MMP9* mRNA expression when MMP14 is inhibited. Treatment of cells with recombinant MMP13 resulted in the opposite pattern—a significant increase in *MMP9* mRNA expression with a corresponding significant decrease in *MMP2* mRNA expression (Figure 5). These expression data are consistent with MMP activity as observed on gelatin zymography (Figure 6); treatment with recombinant MMP13 showed increased MMP9 activity. However, in contrast with our gelatin zymography data, inhibition of MMP13 reduced *MMP2* mRNA expression (Figure 5) while we saw higher MMP2 activity on gel zymography (Figure 6). However, we also saw an increase in total MMP activity (Figure 4) when cells were treated with an MMP13-specific inhibitor. A number of factors may influence differences between MMP mRNA expression and activity [56]. MMP13, also known as collagenase 3, has the highest affinity for type II collagen, cleaving it at a rate up to 5 times higher than all other MMPs and proteases secreted during wounding [57]. MMP13 cleaves collagens and gelatins at the same C-terminal and N-terminal sites into ¾ and ¼ fragments (specific sites Gly906-Leu907 in type II collagen) and additionally at Gly909-Gln910 and Gly910-Ile913 [58,59]. This means that MMP13 may have many functions and needs strict regulation at all levels of expression. Additionally, MMP2 and MMP9 share similar catalytic domains and have the identifying type II hemopexin repeats enabling very specific binding of proteins with matching sites, not just denatured collagen (gelatin) from other MMP collagenases or ECM proteins [4,11,25], therefore there may be some overlap in MMP2 or MMP9 activity that we cannot identify or separate. The observed decrease in *MMP2* mRNA expression implies a potential reduction in MMP2 protein abundance. However, the observed increase in MMP2 activity following MMP13 inhibition on the gel zymogram suggests that suppression of MMP13 may initiate, or release from inhibition, alternative proteolytic cascades capable of activating pro-MMP2 as well as other MMPs [30,43,60], independent of changes in gene expression, as alterations in *MMP2* mRNA expression are not necessarily expected to correlate with changes in MMP2 protein expression or activity within a 24 h period. Although secondary activation of MMPs may occur, there may be post-transcriptional and/or post-translational regulation of MMP2 protein activity; future studies to tease apart the relationship between MMP13 and gelatinase activity of MMP2 and/or MMP9 are needed to identify which mechanism(s) are at play.

Taken together, our data reveal the complex nature of MMP activity in collective cell migration. Given that our experimental system reduces the contribution of the extracellular matrix, our results suggest that MMP activity, regulated by other MMPs, is key in driving collective migration of zebrafish keratocytes. Understanding the interplay among MMPs provides potential therapeutic targets to promote wound healing and may suggest therapeutic avenues to limit metastasis [12,15,61].

## 4. Materials and Methods

### 4.1. Keratocyte Explant Cultures

Wild-type zebrafish (*Danio rerio*) were used for establishing explant cultures for all assays. Zebrafish were anesthetized using 100 µg/mL of tricaine methanesulfonate (MS-222, Sigma-Aldrich, St. Louis, MO, USA) in dechlorinated water until gill movement slowed. Each anesthetized zebrafish was removed from the tricaine solution and placed on ice. Four to eight scales were plucked on each side of the fish, from regions near the caudal fin and superior to the pelvic fin; the pectoral fin and gill regions were avoided. Each plucked scale was placed on a poly-lysine-coated 35 mm glass bottom tissue culture dish with the underside of scale touching the glass surface. Scales were allowed to adhere to the dish for 3–5 min, upon which time 2.0 mL of complete medium (RPMI 1640 plus HEPES, 10% fetal bovine serum albumin, 100 μg/mL kanamycin and 50 μg/mL gentamicin) was added to each dish, defining time 0 for all experiments. No subculturing of cells was performed; each experiment began with explant culture establishment.

### 4.2. DQ Gel Assay

Scales from wild-type zebrafish were plucked and established in explant culture in the same manner as described above. At the time of culture establishment, dishes were left untreated (media alone) or treated with either 10 µM Z-PLG-NHOH or 100 nM NNGH (broad-spectrum MMP inhibitors; Enzo Life Sciences, Farmingdale, NY, USA) and incubated at 28 °C in 5% CO_2_ for 24 h. After incubation, cells were fixed using 4% paraformaldehyde in 1.1× phosphate-buffered saline (PBS) and overlaid with 400 µL of buffer (100 mM NaCl, 100 mM Tris-HCl pH 7.5, 10 mM CaCl_2_, 20 µM ZnCl_2_, 0.05% NP40) containing 20 µg/mL final concentration of a fluorescein-conjugated DQ gelatin substrate, at a (Life Technologies, Carlsbad, CA, USA). Dishes were incubated for an additional 48 h, after which the DQ gelatin was gently washed off the cell sheets, mounted using Prolong Gold Antifade mounting medium with DAPI (Life Technologies) and viewed under 400× magnification using a confocal microscope (Leica, Wetzlar, Germany). The fluorescence intensities at both the leading edge of the cell sheet as well as single-layer cells within the cell sheet (termed follower cells), indicating MMP activity, were measured using ImageJ software (NIH). Average intensities for each treatment were graphed using GraphPad Prism (10.5.0) and analyzed using a two-way ANOVA and Tukey’s multiple comparisons test.

### 4.3. Cell Sheet Migration Assays

Scales from wild-type zebrafish were plucked and established in explant culture in the same manner as described above. At the time of culture establishment, dishes were either left untreated, or were treated with either recombinant, active MMP13 (Enzo Life Sciences), or one of several MMP inhibitors: WAY170523 (MMP13-specific inhibitor; Tocris, Minneapolis, MN, USA), MMP9/13-specific inhibitor I (Santa Cruz Biotechnology), and NSC405020 (MMP14-specific inhibitor; Tocris). Varying doses were used of each compound:Recombinant MMP13: 65 nM and 130 nMMMP13-specific inhibitor (WAY170523): 1 µM, 10 µM, and 100 µMMMP9/13-specific inhibitor I: 90 pM and 900 pMMMP14-specific inhibitor (NSC405020): 10 µM, 100 µM, and 500 µM

For recombinant MMP13, 13 nM demonstrated activity per the manufacturer. We selected four doses, 13 nM plus three incrementally higher doses (26 nM, 65 nM, and 130 nM), to test. As the two lower doses did not yield any effect on keratocyte migration compared with control, we chose to discontinue using these two doses for additional analysis or subsequent experiments. For the MMP inhibitors, we tested the published IC_50_ dose and a dose 10-fold lower and 5-fold or 10-fold higher. The 10-fold IC_50_ dose (9000 pM) for the MMP9/13 inhibitor did not yield any measurable cell sheets and appeared to be toxic to the cells, so was not used beyond initial testing. All treatments were added at time zero and incubated at 28 °C in 5% CO_2_ for 24 h. After a 24 h incubation, each cell sheet was observed under 100× magnification using an Olympus IX40 and images acquired using CellSens imaging software version 1.7 (Olympus, Center Valley, PA, USA). The area of each cell sheet was measured by tracing the perimeter of the cell sheet using ImageJ software (NIH, Bethesda, MD, USA) and graphed using GraphPad Prism 10.5.0. For each assay, medians were analyzed using a Kruskal–Wallis non-parametric one-way ANOVA with Dunn’s multiple comparisons test.

### 4.4. Fluorescence Resonance Energy Transfer (FRET) MMP Assays

Scales from wild-type zebrafish were plucked and established in explant culture in the same manner as described above. At the time of culture establishment, dishes were either left untreated, or were treated with 65 nM recombinant MMP13, 170 nM of an MMP13 inhibitor (WAY170523; Enzo Life Sciences), 250 µM of an MMP14-specific inhibitor (NSC405020), or 10 µM of a broad-spectrum (total) MMP inhibitor (Z-PLG-NHOH) and incubated at 28 °C in 5% CO_2_ for 24 h. After incubation, culture supernatants were collected and concentrated 10-fold using 10K MWCO spin filters. MMP13-specific and total MMP activity was measured using a 5-FAM/QXL™ 520 FRET Assays kit, separately (AnaSpec, Inc., Fremont, CA, USA), following the manufacturer’s recommended protocol. Briefly, 50 µL of each culture supernatant was incubated with 1 mM aminophenylmercuric acetate (APMA) at 37 °C for 40 min (MMP13-specific assay) or 60 min (total MMP assay) to activate MMPs within the supernatants. After incubation, the MMP13-specific substrate solution (AnaSpec, Inc.) was diluted 1:100 and 50 μL was added to each culture supernatant and placed into a well within a 96-well plate. The same procedure was performed for all supernatants in triplicate. This process was repeated, also in triplicate, to assess total MMP activity using a total MMP activity kit (AnaSpec, Inc.). The fluorescence signal was measured in a kinetic assay every 5 min for 60 min using a Biotek Cytation3 plate reader (Biotek, Winooski, VT, USA). Positive (MMP13 and total MMP substrates) and negative (media, buffer, and no APMA) controls were also performed. Relative MMP activities over time, represented by relative fluorescent units, were calculated using Excel (Microsoft 2011). Results from each FRET assay were analyzed using a two-way ANOVA with Dunnett’s multiple comparisons test and graphed using GraphPad Prism 10.5.0.

### 4.5. EdU Proliferation Assay

Zebrafish explant cultures were set-up as previously described. At the time of culture establishment, dishes were either untreated or treated with varying doses of recombinant MMP13, an MMP13-specific inhibitor, and an MMP14-specific inhibitor as described in Section 4.3 above, labeled with EdU nucleotide (5-Ethyl-2′-Deoxyuridine) at a concentration of 10 μM per dish, and incubated at 28 °C in 5% CO_2_ for 24 h. Cells were fixed by adding 1.0 mL of 3.7% formaldehyde in PBS, washed in 3% BSA-PBS solutions and permeabilized using 0.5% Triton X-100 in 1× PBS. After fixation, 500 µL of the EdU Click-It^©^ reaction solution (Invitrogen: Click-iT buffer, CuSO_4_, Alexa Fluor azide, and additive) was added per dish; each glass coverslip was subsequently mounted using Prolong Gold Antifade mounting medium with DAPI (Life Technologies). Cell sheets in each dish were imaged using an Apotome microscope (Zeiss, Oberkochen, Germany) under 1000× magnification. For each image, the total number of cells (DAPI stained) within each cell sheet as well as the number of proliferating cells (EdU stained) were counted using CellSens Dimension software (Olympus) and the percentage of cells relative to the total number of cells was calculated using Excel (Microsoft 2011). Statistical significance among treatments was determined using a Kruskal–Wallis non-parametric one-way ANOVA with Dunn’s multiple comparisons test and graphed using GraphPad Prism 10.5.0.

### 4.6. RNA Isolation, mRNA Amplification and cDNA Synthesis

RNA was harvested using RNApure (GenHunter, Nashville, TN, USA) from untreated cells and cells treated with one of the following: 65 nM recombinant MMP13, 170 nM MMP13-specific inhibitor, 250 µM MMP14-specific inhibitor, or 10 µM broad-spectrum (total) MMP inhibitor. Cultures were initially set up during the cell sheet migration assays as described in Section 4.3. Isolated RNA was stored at −80 °C for no more than 72 h before being thawed on ice. RNA from each experimental condition was purified using an RNeasy MinElute Cleanup kit (Qiagen, Hilden, Germany), quantitated using a Nanodrop spectrophotometer, and sent to the Arizona State University Core Facilities for Bioanalysis (Agilent 2000 Bio analyzer, Santa Clara, CA, USA); all RIN values were between 7.2 and 9.1. 2.0 µg of each RNA sample was treated with DNase I for 30 min at 37 °C, then heat-inactivated for 10 min at 75 °C. DNase-treated RNA was converted to cDNA using SuperScript II and pdN6 random hexamer primers (Life Technologies). Each cDNA sample was either immediately used in a qPCR reaction or stored at −80 °C until use.

### 4.7. Quantitative Real-Time PCR (qPCR)

Custom primers for four target genes (*MMP2*, *MMP9*, *MMP13a* and *MMP14a*) and two endogenous reference genes (*β-actin* and *EF1α*) were synthesized by IDT^®^ (Integrated DNA Technologies, Inc., Coralville, IA, USA) and rehydrated to 100 µM using 1× Tris-EDTA buffer. Sequences are listed in Table 1. Each primer pair was validated for efficiency; *β-actin* was the endogenous reference gene chosen as this primer pair amplified DNA with 100% efficiency.

For each cDNA sample, a qPCR reaction was performed in duplicate for each target and reference gene. All reactions were performed in 96-well plates (MicroAmp, Warsaw, Poland) and amplification was detected using SYBR Green (SA Biosciences, Frederick, MD, USA) and a Bio-Rad CFX96 Real time PCR system. After each qPCR reaction, the average ΔCt was calculated for each target and endogenous reference gene for each treatment condition. A relative standard curve approach, involving the normalization of target gene expression to the endogenous reference gene, was used for all experiments. To determine relative expression levels of each target gene, the ΔΔCt method was performed. Fold changes in gene expression were calculated using Excel (Microsoft 2011) and analyzed and graphed in GraphPad Prism (10.5.0) using a two-way ANOVA and Dunnett’s multiple comparisons test.

### 4.8. Gelatin Zymography

Wild-type: zebrafish explant cultures were established in six well dishes, 10–12 scales per well, with 400 μL/well of serum free culture media (GIBCO serum free keratocyte medium, 2% FBS (Fetal Bovine Serum, 50 µg/mL gentamicin, 100 µg/mL kanamycin, 25 mM Hepes) added to each well in the same manner as described above. At the time of culture establishment, dishes were either untreated (control) or treated with 65 nM of recombinant MMP13, 170 nM of an MMP13-specific inhibitor (WAY170523; Enzo Life Sciences), 250 µM of an MMP14-specific inhibitor (NSC405020; Tocris), or 10 µM of a broad-spectrum (total) MMP inhibitor (Z-PLG-NHOH; Tocris) and incubated at 28 °C in 5% CO_2_ for 24 h. After incubation, conditioned medium was pooled for each treatment and concentrated 10-fold using 10K MWCO spin filters. A BCA assay was performed on all samples for total protein concentration determination. Gel zymography was performed following the protocol outlined by Frankowski et al. [62]. Briefly, normalized total protein of 100 μg per sample was loaded onto a 10% resolving gel containing 0.2% gelatin and SDS with a 5% stacking gel. Samples were mixed with 5× non-reducing sample buffer, pH 6.8 (0.313 M Tris–HCl, pH 6.8, 10% SDS, 50% glycerol, 0.05% bromophenol blue) and incubated for 10 min at room temperature. The gel was pre-run at 100 volts for 30 min at which time samples were loaded into the wells and the gel ran at 40 volts for 2.5 hr followed by 85 volts for 16 h at 4 °C using precooled Tris/glycine/SDS running buffer. After electrophoresis, the gel was carefully removed and washed 3× for 15 min each with 60 mL of gel washing buffer (0.025% Triton X-100 solution in sterile diH_2_O) under agitation. Following washing, the gel was incubated with gel development buffer (50 mM Tris HCl, 0.15 M NaCl, 10 mM CaCl_2_, and 0.05% NaN_3_) at 37 °C for 24 h. The gel was stained using 0.05% Coomassie blue staining solution (0.05% Coomassie brilliant blue R-250, 10% acetic acid, 25% methanol) for 1 h at room temperature with constant agitation and then destained with destaining solution (4% methanol, 16% acetic acid) for 3–5 h, until clear bands were revealed. The gel was imaged using a Bio-rad Chemi-Doc™ XRS plus gel imaging system (Hercules, CA, USA). Quantification of each band in each lane was measured horizontally between samples using ImageJ Fiji 2.16.0 and the area of each band was normalized to the corresponding area of the bovine serum albumin (BSA) band in each lane. Data were analyzed in Excel (Microsoft 2011) and graphed using GraphPad Prism (10.5.0), and statistical analysis among conditions was performed using a two-way ANOVA and Dunnett’s multiple comparisons test.

## Figures and Tables

**Figure 1 ijms-26-11192-f001:**
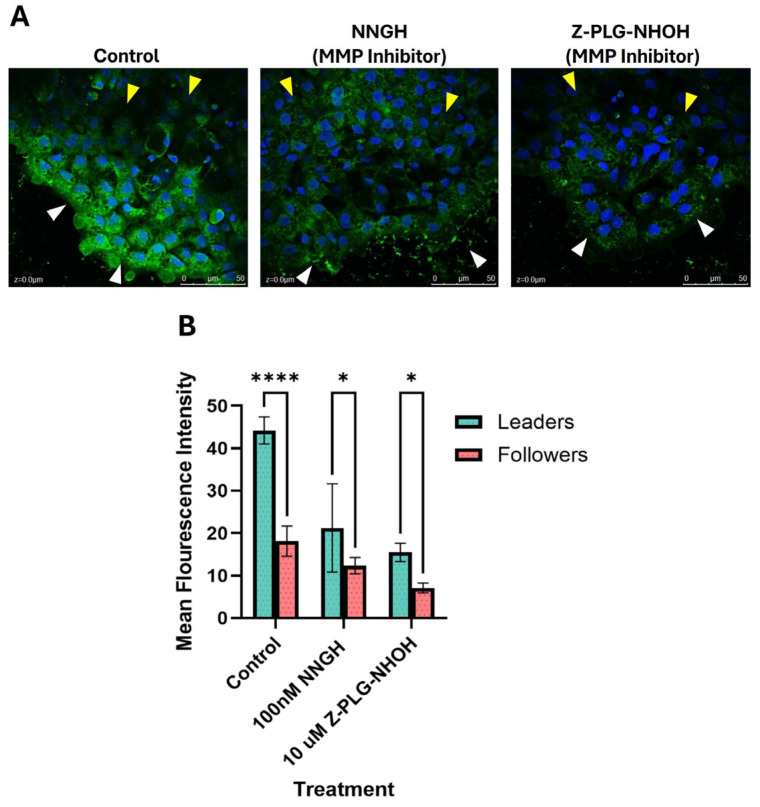
MMP activity in cells at the leading edge of a collectively migrating sheet (leaders) versus cells within the sheet, away from the leading edge (followers). Untreated cells and cells treated with a broad-spectrum MMP inhibitor (either NNGH or Z-PLG-NHOH) were assessed for MMP activity using a fluorescent DQ gel assay. (**A**) Representative DQ gel images showing MMP expression (green) in the leader cells (white arrowheads) and follower cells (yellow arrowheads); nuclei were stained with DAPI (blue). (**B**) MMP activity was significantly higher in untreated cells at the leading edge of the migrating sheet compared to follower cells, with a sample size (n) of 8 per treatment. Key: * denotes *p* < 0.05 and **** denotes *p* < 0.0001.

**Figure 2 ijms-26-11192-f002:**
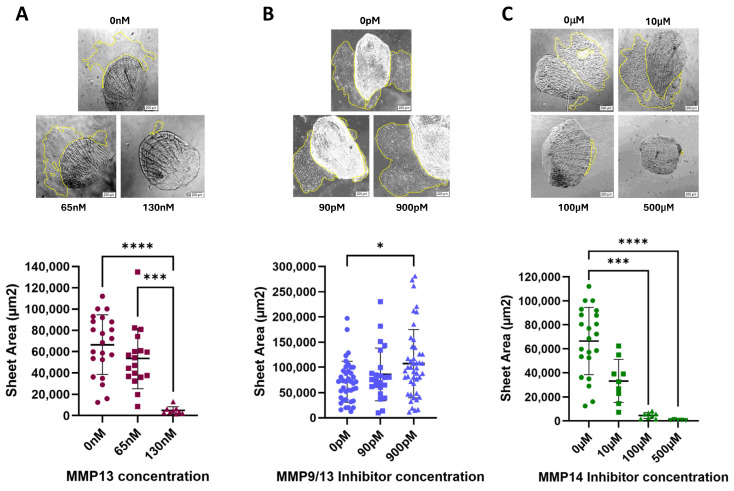
Keratocytes were plated and treated with varying doses of (**A**) recombinant MMP13; sample size (n) ranging from 9–23, (**B**) the MMP9/13-specific inhibitor I (IC_50_ of 900 pM); sample size (n) ranging from 23–45 or (**C**) the MMP14-specific inhibitor NSC405020 (IC_50_ of 100 µM); sample size (n) ranging from 5–22. After 24 h, the area of each migrating cell sheet was measured. Representative images of cell sheets for each dose of MMP13 or MMP inhibitor tested after 24 h of treatment are shown, with yellow dotted lines outlining the area of the cell sheet; all images are at 100× magnification. The horizontal line represents the median cell sheet area. Key: * denotes *p* < 0.05, *** denotes *p* < 0.001, and **** denotes *p* < 0.0001.

**Figure 3 ijms-26-11192-f003:**
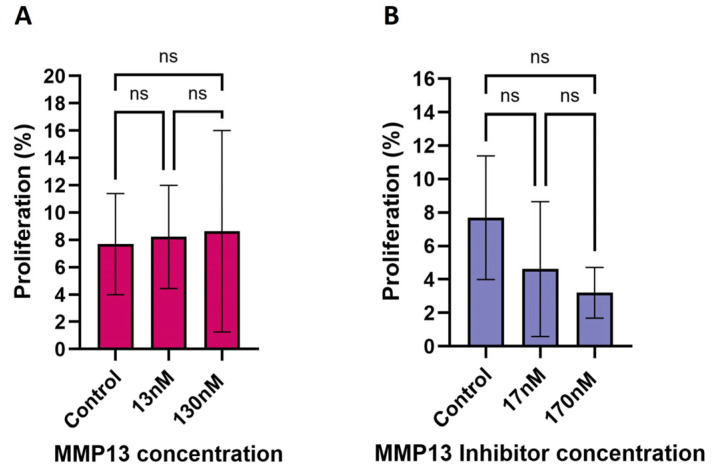
EdU assay showing the effect of recombinant MMP13 and MMP13-specific inhibition on the rate of keratocyte proliferation after 24 h of treatment. (**A**) Treatment with recombinant MMP13 (13 nM & 130 nM) had no significant (ns) effect on proliferation compared to untreated cells. (**B**) Treatment with the MMP13-specific inhibitor WAY170523 (17 nM & 170 nM) also had no significant effect on proliferation. Data are mean ± SD with a sample size (n) ranging from 7 to 12 per treatment.

**Figure 4 ijms-26-11192-f004:**
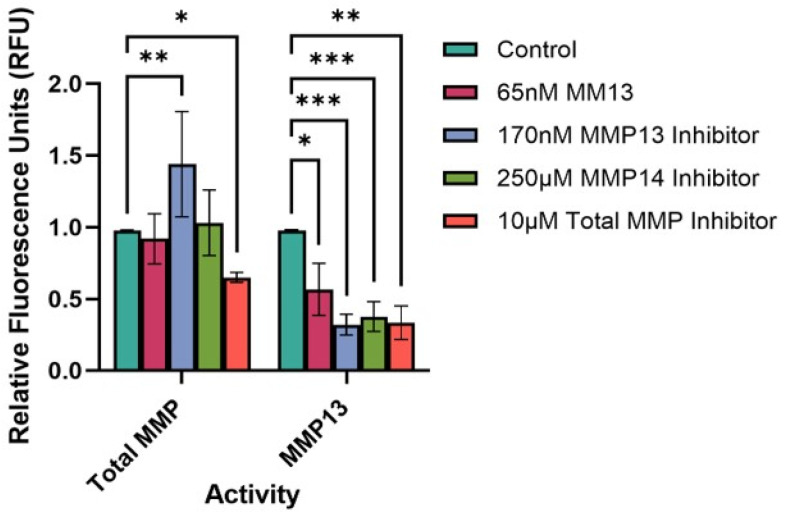
FRET assay to determine the total MMP and MMP13-specific activity in keratocyte culture supernatants. Cells were either untreated (control) or treated with recombinant MMP13, or an MMP inhibitor (MMP13, MMP14, or broad-spectrum [total] MMP inhibitor) for 24 h, upon which a FRET assay was performed to assess MMP activity. Bars represent mean ± SD with a sample size (n) of 3–4 per treatment. Key: * denotes *p* < 0.05, ** denotes *p* < 0.01, and *** denotes *p* < 0.001.

**Figure 5 ijms-26-11192-f005:**
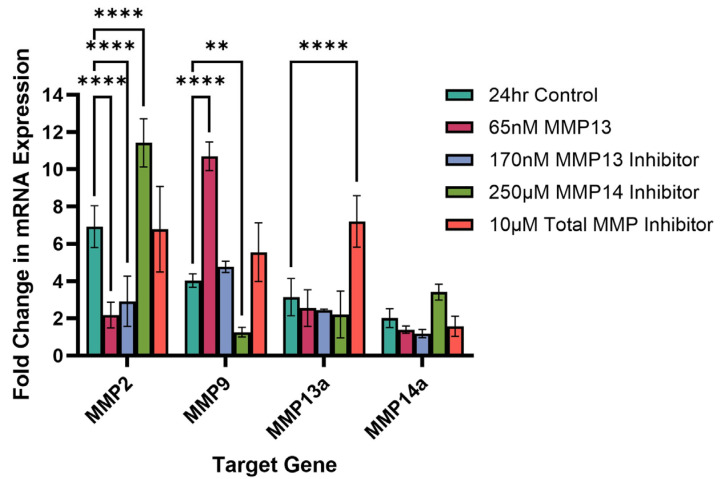
Quantitative PCR measuring the fold change in mRNA expression of *MMP2*, *MMP9*, *MMP13a*, and *MMP14a* following treatment with recombinant MMP13 or an MMP inhibitor for 24 h. Untreated cells were used as control. Bars represent mean ± SD with a sample size (n) of 3 per treatment. Key: ** denotes *p* < 0.01 and **** denotes *p* < 0.001.

**Figure 6 ijms-26-11192-f006:**
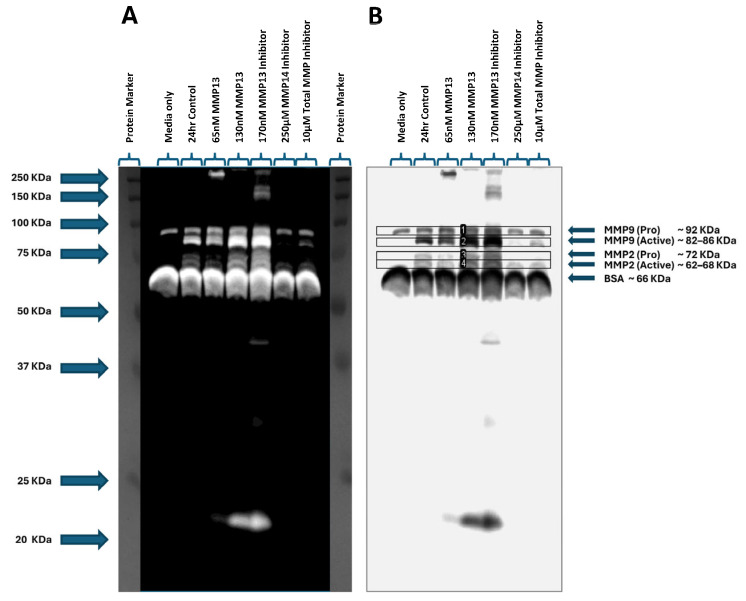
Gelatin zymography assessing gelatinase activity in keratocyte cell culture supernatants. Media only and untreated cells were used as controls. Supernatants from cells treated with 65 nM or 130 nM recombinant MMP13 or treated with an MMP13 inhibitor, an MMP14 inhibitor, or a broad-spectrum (total) MMP inhibitor were loaded and the molecular weights assessed on a zymogram gel. (**A**) Positive staining of the zymogram gel aligning each band with the corresponding molecular weight protein marker. (**B**) Negative staining of the zymogram gel was used to quantitate each band within the indicated boxes (labeled 1–4). (**C**) Graph showing increased MMP2 gelatinase activity in samples treated with an MMP13 inhibitor compared to untreated cells (24 h control) (*p* < 0.01). (**D**) Graph depicting an increase in MMP9 gelatinase activity following treatment with 130 nM of recombinant MMP13 (*p* < 0.01) or an MMP13 inhibitor (*p* < 0.05), and a decrease in MMP9 activity after treatment with an MMP14 inhibitor (*p* < 0.05) compared to untreated cells. Data are represented as mean ± SD with a sample size (n) of 3 per treatment. Key: * denotes *p* < 0.05 and ** denotes *p* < 0.01.

**Table 1 ijms-26-11192-t001:** Forward and reverse primer sequences used for qPCR amplification of four MMP (target) genes and 2 endogenous reference genes.

**Gene**	**Primer Sequence**
*MMP2*	F: GAG CTC TCA TGG CTC CTA TCT A R: TGG CTT GTC TGT TGG TTC TC
*MMP9*	F: TTT GCC CTG ATC GTG GAT AC R: GGG AAA CCC TCC ACG TAT TT
*MMP13a*	F: CTG GCC TGA GAT TCC AGA TAA C R: CAT AGA GAG CCC AAA CCT TCT C
*MMP14a*	F: GAC AAA GAA GTG AGA CCA GAG G R: TTT CTG CAT GGC CGA GAT AG
*β-actin*	F: GCA AAG GGA GGT AGT TGT CTA A R: GAG GAG GGC AAA GTG GTA AA
*Elongation Factor 1α (EF1α)*	F: ATG CCC TTG ATG CCA TTC T R: CCC ACA GGT ACA GTT CCA ATA C

## Data Availability

The original contributions presented in this study are included in the article/Appendix A. Further inquiries can be directed to the corresponding author.

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
