# Peer review of "The Coordinated Interplay Between MMP13 and Pro-Migratory MMPs in Collective Cell Migration of Zebrafish Keratocytes"

_ijms, 2025, doi:10.3390/ijms262211192_

Round 1
Reviewer 1 Report
Comments and Suggestions for Authors
The authors investigated the migration of zebrafish keratocytes regulated by MMP13 and MMP14 or their inhibitors, as well as whether MMP13 and MMP14 regulated other MMPs. Here are my comments:
1: Keratocyte Explant Cultures: Details are missing. For example, how many days were required to obtain cells ready for use? Was any subculture involved? Please include representative images of the cells at the stage when they were used.
2: Figure 1:
- A) Move Supplemental Figure 1 to Figure 1.
- B) How were the concentrations of MMP inhibitors determined?
3: Figure 2:
- A) Add representative images.
- B) Was mitomycin C used in the migration assay? If not, the experiment should be repeated with mitomycin C to rule out the effect of cell proliferation.
4: Figure 4: Please explain how “treatment of cells with 65 nM MMP13 also significantly reduced MMP13 activity in culture supernatants after 24 hours” can be explained.
Author Response
The authors investigated the migration of zebrafish keratocytes regulated by MMP13 and MMP14 or their inhibitors, as well as whether MMP13 and MMP14 regulated other MMPs. Here are my comments:
1: Keratocyte Explant Cultures: Details are missing. For example, how many days were required to obtain cells ready for use? Was any subculture involved? Please include representative images of the cells at the stage when they were used.
- Thank you for your comment, as it underscores that a more complete description of the primary explant culture system utilized in this work is needed. We have edited section 4.1, lines 376-377, to clarify. Keratocyte explant cultures are established at the time of removing a scale with cells from an anesthetized zebrafish, then plating (adhering) the scale containing cells into a glass-bottom culture dish. Media and any treatment were added at time 0, and the dishes were incubated for 24 hours. The area of cellular migration was measured after 24 hours. No subculturing of cells was done. Representative images of cell sheets after 24 hours have been added to Figure 2. At time 0, which is immediately after plucking a scale and adding media, the images would only show a scale in the dish; cells are under the scale but often hard to visualize until they migrate out from under the scale.
2: Figure 1:
- A) Move Supplemental Figure 1 to Figure 1.
- We thank you for this suggestion – we considered doing so before submission, so we were pleased with your request to move the images in supplemental Figure 1 to Figure 1. We removed the reference to the supplemental figure in the text, Section 2.1 (lines 97 and 99) and updated the figure legend (lines 101-106). The original Supplemental Figures 2 & 3 are now renamed Supplemental Figures 1 & 2 and edited in the text (lines 134-135, 153, 528-532, and 548).
- B) How were the concentrations of MMP inhibitors determined?
- Thank you for this suggestion and we do apologize for not properly addressing to describe how these doses were selected. The selection of doses was based on the published IC50 We edited the text, lines 406-413, to provide this information.
3: Figure 2:
- A) Add representative images.
- We thank you for this suggestion and agree that this enhances the figure. We have added representative sheet images for each dose tested and updated the figure legend (lines 125-127) accordingly. We added an image of an untreated cell sheet for Figures 2A and 2B, but did not for Figure 2C for space. Data collected for untreated cells were used as controls against all treatments.
- B) Was mitomycin C used in the migration assay? If not, the experiment should be repeated with mitomycin C to rule out the effect of cell proliferation.
- We did not use mitomycin C in our migration assays. Our experimental design was based on our previous experience with this explant culture system. We have found that our sheet assays utilizing a single treatment allow us to minimize potential toxicity or off-target effects. Instead of using mitomycin C in combination with treatments, we took a serial approach to address the relative contributions of migration and proliferation. We first measured the sheet area after treatments, and then we addressed the contribution of proliferation using an EdU assay. We did not expect to see much proliferation within the 24 hour time frame of the experiment, and our EdU assays confirmed that there was no change in proliferation regardless of treatment. Therefore, we can exclude proliferation as an explanation to the increases in sheet areas measured with treatments.
4: Figure 4: Please explain how “treatment of cells with 65 nM MMP13 also significantly reduced MMP13 activity in culture supernatants after 24 hours” can be explained.
- We thank you for this comment. We were perplexed by this finding, however treatment with MMP13 did not affect total MMP activity compared to untreated cells. We will be exploring this finding to determine if there is any biological significance to this result. From Figure 6, we did not observe any difference in MMP2 or MMP9 activity when cells were treated with 65 nM MMP13, but did see increased activity at the highest dose (130 nM) tested. We edited the text (lines 312-316) to provide some additional explanation.
Reviewer 2 Report
Comments and Suggestions for Authors
- Inconsistency Between mRNA and Protein Activity Data for MMP2:A critical point of confusion arises from the data on MMP2. Inhibition of MMP13 led to:
A decrease in MMP2 mRNA expression (Figure 5).
An increase in MMP2 gelatinase activity (Figure 6C). This discrepancy is noted by the authors but explained away with a brief, speculative paragraph. This needs a more rigorous exploration. Is this a post-translational activation phenomenon? Is the protein more stable or more efficiently activated? Simple Western blot analysis for MMP2 protein levels (pro and active forms) in these conditions would be highly informative to resolve this contradiction and is strongly recommended.
- Figures 6 A and B need to increase their resolution
- The zymography data in Figure 6 is central to the argument, yet the methodology states "n = 2 per treatment." This is an unacceptably low sample size for robust statistical analysis. The p-values presented (<0.05, <0.001) are not credible with n=2. This experiment must be repeated with a biologically appropriate n (at least n=3-4 independent replicates) to validate these crucial findings.
- The data for MMP14's effect on proliferation is incomplete, as higher doses prevented sheet growth. The authors' hypothesis that it does not affect proliferation is just that—a hypothesis. This should be stated more cautiously, acknowledging the limitation of the data.
- There are minor typos, e.g., "contradict was has been" should be "contradict what has been" (Page 2), and "MMP-14" is sometimes written with a hyphen and sometimes without. A thorough proofread is recommended.
Author Response
- Inconsistency Between mRNA and Protein Activity Data for MMP2:A critical point of confusion arises from the data on MMP2. Inhibition of MMP13 led to:
A decrease in MMP2 mRNA expression (Figure 5).
An increase in MMP2 gelatinase activity (Figure 6C). This discrepancy is noted by the authors but explained away with a brief, speculative paragraph. This needs a more rigorous exploration. Is this a post-translational activation phenomenon? Is the protein more stable or more efficiently activated? Simple Western blot analysis for MMP2 protein levels (pro and active forms) in these conditions would be highly informative to resolve this contradiction and is strongly recommended.
- We thank you for this comment. Fundamentally, we were interested in protein activity and recognize that mRNA expression does not always correlate with protein levels and/or activity. However, we do maintain that the zymography data does assess activity, which is fundamental to the biological role of MMPs. So, while we agree that performing a western blot may provide some explanation, we are unable to directly address this concern for two reasons: 1. We were unable to find a zebrafish-specific antibody that would accurately and definitively assess protein concentration between the pro and active forms of MMP2 in our system, and 2. Given the 10-day turnaround time we were given to provide a revised manuscript, performing these additional experiments are not feasible to do and are outside the scope of this current manuscript. We are highly interested in exploring this further and several additional experiments would likely be required to address this adequately. We have provided some additional explanation in the discussion (lines 346-357).
- Figures 6 A and B need to increase their resolution
- Thank you for the suggestion. We have done the best we can to increase the resolution of Figures 6A and 6B, and hope that the updated images are satisfactory.
- The zymography data in Figure 6 is central to the argument, yet the methodology states "n = 2 per treatment." This is an unacceptably low sample size for robust statistical analysis. The p-values presented (<0.05, <0.001) are not credible with n=2. This experiment must be repeated with a biologically appropriate n (at least n=3-4 independent replicates) to validate these crucial findings.
- We thank you for your comment, and we agree that having an additional replicate was needed. We analyzed one additional replicate, resulting in an n=3. While the p-values changed, the data still show the same pattern of statistically significant differences as we reported previously, except for the control vs. the broad-spectrum MMP inhibitor. Even though the data shows a similar pattern, the p-value increased above 0.05 for this comparison when we added in data from a third replicate. We have updated the text, figures, and p-values accordingly (Figures 6C and 6D, lines 202-206, and lines 219-223).
- The data for MMP14's effect on proliferation is incomplete, as higher doses prevented sheet growth. The authors' hypothesis that it does not affect proliferation is just that—a hypothesis. This should be stated more cautiously, acknowledging the limitation of the data.
- We thank the review for this comment. We have added representative cell sheet images to Figure 2 (based on another reviewer suggestion), which show that treatment with 500 mM MMP14 does not yield visible cell sheets. Therefore, regardless of any effect on proliferation, this dose does not contribute to a collective cell migration process. We have edited the text accordingly (lines 288-292) to increase clarity.
- There are minor typos, e.g., "contradict was has been" should be "contradict what has been" (Page 2), and "MMP-14" is sometimes written with a hyphen and sometimes without. A thorough proofread is recommended.
- Thank you for this comment and we apologize for these errors. We have performed a thorough proofread to correct grammatical and typographical errors throughout the manuscript.
Round 2
Reviewer 1 Report
Comments and Suggestions for Authors
The authors made efforts to improve the manuscript. We have three additional minor suggestions for Figure 2:
- Use dotted lines to mark the cell sheet migration areas, so the readers can easily identify them.
- The image of the control is missing in C. Please add it.
- Add scale bars.
Author Response
The authors made efforts to improve the manuscript. We have three additional minor suggestions for Figure 2:
Comment 1: Use dotted lines to mark the cell sheet migration areas, so the readers can easily identify them.
Response 1: We thank this reviewer for this excellent suggestion. We have added in yellow dotted lines outlining each cell sheet at the top of Figure 2.
Comment 2: The image of the control is missing in C. Please add it.
Response 2: We thank this reviewer for this suggestion. We agree that having a representative image for each treatment is preferable, so we have added in a control image for Figure 2C.
Comment 3: Add scale bars.
Response 3: We agree that a scale bar is a good addition and we have added one to each image. We noticed that we incorrectly listed the magnification at 400X; all images were taken at 100X magnification, and this has been corrected.
Reviewer 2 Report
Comments and Suggestions for Authors
The authors have made the requierd modifications, so Accept
Author Response
Comment 1: The authors have made the requierd modifications, so Accept.
Response 1: We thank the reviewer for carefully reading through our manuscript a second time and are pleased that the edits we made are satisfactory.